# The Polyvalent Role of NF90 in RNA Biology

**DOI:** 10.3390/ijms232113584

**Published:** 2022-11-05

**Authors:** Giuseppa Grasso, Rosemary Kiernan

**Affiliations:** UMR9002 CNRS-UM, Institut de Génétique Humaine-Université de Montpellier, Gene Regulation Lab, 34396 Montpellier, France

**Keywords:** NF90, miRNA, RISC, posttranscriptional regulation, gene regulation, dsRBP

## Abstract

Double-stranded RNA-binding proteins (dsRBPs) are major players in the regulation of gene expression patterns. Among them, Nuclear Factor 90 (NF90) has a plethora of well-known functions in viral infection, transcription, and translation as well as RNA stability and degradation. In addition, NF90 has been identified as a regulator of microRNA (miRNA) maturation by competing with Microprocessor for the binding of pri-miRNAs in the nucleus. NF90 was recently shown to control the biogenesis of a subset of human miRNAs, which ultimately influences, not only the abundance, but also the expression of the host gene and the fate of the mRNA target repertoire. Moreover, recent evidence suggests that NF90 is also involved in RNA-Induced Silencing Complex (RISC)-mediated silencing by binding to target mRNAs and controlling their translation and degradation. Here, we review the many, and growing, functions of NF90 in RNA biology, with a focus on the miRNA pathway and RISC-mediated gene silencing.

## 1. Introduction

Appropriate and controlled gene expression is essential for every living organism. Deregulation of this process can have profound effects and often be responsible for diseases such as cancer and genetic disorder [1,2,3]. While specific gene expression can be achieved at the step of gene transcription, numerous regulatory mechanisms contribute to gene expression patterns post-transcriptionally [2].

RNA-binding proteins (RBPs) are known to contribute to posttranscriptional gene regulation, mediating mRNA splicing, translation, and turnover [4,5]. Among them, Nuclear Factor 90 (NF90) is a double-stranded RNA-binding protein that is part of a family of proteins produced from the Interleukin Enhancer Binding Factor 3 (*ILF3*) gene [6]. *ILF3* is localized on human chromosome 19 and contains 21 exons, giving rise to at least five distinct transcripts that are generated by alternative splicing [7]. The most abundant isoforms produced from the *ILF3* gene are NF90 and Nuclear Factor 110 (NF110), which are ubiquitous and generally abundant proteins with an apparent molecular mass of 90 kDa and 110 kDa, respectively [6,7,8,9,10,11].

The main protein partner of NF90 and NF110 is Nuclear Factor 45 (NF45), which is transcribed from the *ILF2* gene. Heterodimerization of NF90/NF110 with NF45 is important for the stabilization and function of the complex. In particular, it was shown that the binding of NF45 to NF90 leads to thermodynamic stabilization and improves the RNA-binding ability of NF90, enhancing its affinity for RNA substrates [12]. In most cell types, NF90 and NF45 are highly abundant. Moreover NF90 is often tightly complexed with NF45, predominantly in the nucleus, in an RNA-independent manner [13,14]. However, NF90 and NF45 have both been shown to shuttle between the nucleus and the cytoplasm, according to their phosphorylation status [15] and as a result of several stimuli [16,17,18,19]. In addition, NF90 can associate with Exportin-5 (XPO5), which promotes its nuclear export in an RNA-dependent manner [20]. NF90/NF45 is surprisingly versatile, being involved in a variety of RNA-dependent processes in both the nucleus and the cytoplasm. It is implicated in transcription [21], splicing [22], post-transcriptional modification, translation, and mRNA stability [23,24,25]. To add to this list, functions in circular RNA formation, miRNA processing, and RNA decay pathways have more recently been described [16,26,27]. Here, we review the polyvalent role of NF90 in RNA biology.

## 2. *ILF3* Gene Products: NF90 and NF110

### 2.1. ILF3 Gene and Transcripts

NF90 and its longer isoform NF110 are produced from the *ILF3* gene, localized in human chromosome 19p13 (Figure 1). The two isoforms share 17 out of 21 exons. However, the C-terminus of NF90 is encoded by exon 18, which is lacking in NF110, while that of NF110 is coded by exon 21 [7,10].

The human *ILF3* gene is a substrate for several observed alternative splicing events. For instance, exon 3 contains an alternative splicing site generating, for both NF90 and NF110, a long isoform containing exon 3 and a short one that lacks it [28]. Another alternative splicing event occurs at the 3′ splicing site between intron 13 and exon 14. If the competing 3′ splice site is recognized instead of the canonical one, four additional amino acids (NVKQ) are translated, encoding for NF90a/b and NF110a/b, where b forms contain the insert [7,29]. More recently, another splicing event within exon 18 was identified in human cancer [30].

### 2.2. Domain Structure of NF90, NF110, and NF45

Human NF90 and the longer isoform, NF110, share the same N-terminal and central regions but differ at the C-terminus. The region common to NF90 and NF110 contains a zinc finger-associated domain (DZF), a nuclear localization signal (NLS), two double stranded RNA-binding motifs (dsRBMs), and a RGG motif (Figure 1). The RGG motif typically consists of several copies or arginine and glycine repeats and they interact with single-stranded RNAs. The RGG motif, together with the two dsRBMs, cooperate to dynamically determine the NF90 RNA-binding ability [31]. The two tandem dsRBMs are separated by a natively unstructured segment and they participate in RNA binding in two different ways. Nevertheless, both dsRBMs contribute to the binding of the same molecule simultaneously and co-operatively. However, while dsRBM2 was shown to be the major determinant in the interaction with double-stranded RNAs (dsRNAs), the involvement of dsRBM1 is minimal.

The DZF domain is also conserved in NF45 and is responsible for dimerization and consequent stabilization of the heterodimer [14]. However, NF45 does not bind RNA efficiently and does not participate in RNA binding by the heterodimer [12]. It was shown that NF45 binding to NF90 is able to drive tertiary structural changes that result in an enhanced interplay between NF90 dsRBMs [12]. Another similarity to RNA editing enzymes comes from the DZF domains of NF90 and NF45, which present a nucleotidyltransferase fold, typical of the RNA modifying enzymes. However, both NF90 and NF45 have lost the critical catalytic residue and, therefore, are not functional RNA editing enzymes [14].

### 2.3. Post Translational Modifications and Regulation of NF90 Expression

NF90 and NF110 contain multiple residues that can be phosphorylated in both the common region and in the isoform-specific regions (Figure 1). Phosphorylation is one of the main translational modifications that can be found on NF90/NF110 and it has been shown to efficiently regulate NF90 compartmentalization between the nucleus and cytoplasm [15,23]. Upon T-cell activation, AKT phosphorylates NF90 on Ser^647^, leading to translocation of NF90 from the nucleus to the cytoplasm [32]. In addition, cyclin-dependent kinase (CDK) 2 was shown to interact with NF90 and phosphorylate Ser^382^, inducing nuclear export of NF90 [23]. The interferon-inducible dsRNA-dependent kinase, Protein Kinase R (PKR), phosphorylates NF90 and NF110 on Thr^188^ and Thr^315^, leading to their nuclear export [33]. Moreover, specific phosphorylation events occurring during mitosis such as phosphorylation at Ser^482^ increase the stability of NF90 [34]. Aside from phosphorylation, asymmetric dimethylation of an arginine in the RGG motif has also been described for NF90. Arginine methylation in the RGG motif of RBPs is usually associated with RNA metabolism, but its role in NF90 function is still unclear [35].

Regarding the regulation of NF90 expression, it was shown that long non-coding RNA (lncRNA) Low Expression in Tumor (lncRNA-LET) destabilizes NF90 by changing its conformation and exposing ubiquitination sites that promote its proteolytic degradation [36]. Recently, it was shown that, in hepatocellular carcinoma, where high NF90 expression is associated with a poor prognosis, ubiquitin-specific protease 11 (USP11) promotes NF90 de-ubiquitination, thereby stabilizing it [37].

## 3. RNA Binding Mode of NF90

NF90 appears to recognize target RNAs based on both structure and sequence. In particular, NF90 binding to the adenovirus-expressed Viral Associated 1 (VA1) RNA did not show any specificity for nucleotide sequence, upon extensive mutational analysis. Instead, the structure of the RNA determined NF90 binding, with a requirement for a minihelix-like structure [20,27]. Furthermore, it was more recently shown that the interaction of NF90 dsRBMs with the target RNA molecule largely recalls the RNA binding mode of Adenosine Deaminase RNA Specific 2 (ADAR2). This analogy suggests that, like for ADAR2, the recognition of a base-specific G-X_10_-A motif in the minor groove could contribute to the interaction of NF90 to target dsRNA, further stabilizing it [38]. In addition, it was shown that NF90 could be involved in mRNA translational regulation by binding a 25- to 30-nucleotide long AU-rich signature motif in the 3′ UTR, called NF90m [39]. However, it is not known whether RNA structure may also be implicated in binding to NF90m. Taken together, these findings suggest that NF90 might recognize target dsRNAs based on either structure, sequence motif, or both, possibly depending on the target itself.

## 4. Functions of NF90

NF90 is a polyvalent factor that, since its discovery, has been linked to a variety of functions such as transcriptional and translational regulation, viral replication, and miRNA biogenesis (Figure 2). Moreover, deregulation of NF90 was observed for several diseases such as cancer and muscular atrophy, and it was implicated in the immune response, particularly against viruses [22,29,40]. More recently, NF90 was also shown to regulate embryonic stem cell pluripotency and differentiation [41].

In recent years, NF90 has been found to be differentially expressed in several types of cancers such as ovarian, breast, cervical, hepatocellular and nasopharyngeal carcinomas, leukemia, and bladder cancer [22,37,42,43,44]. While the impact of NF90 on cancer proliferation, progression, metastasis, and drug resistance is evident, its role in these processes is divergent and may depend on the cancer type. For instance, NF90 was shown to be a strong tumor suppressor for ovarian carcinoma while promoting proliferation and metastasis in hepatocellular carcinoma (HCC) [22,37].

### 4.1. NF90 in Transcriptional Regulation

Although NF90 does not contain a known DNA-binding motif, evidence strongly suggests that it could have a similar function to canonical DNA-binding proteins. NF90/NF45 was originally described as a DNA-binding complex, acting as a transcription factor for the cytokine, interleukin 2 (IL2), during T-cell activation. In particular, it was shown that NF90/NF45 binds the antigen recognition response element 2 (ARRE-2) present in the *IL2* promoter and enhances its transcription [45,46]. It was subsequently demonstrated that the interaction between the complex and target DNA is indirect, being mediated by several protein partners such as eukaryotic Translation Initiation Factor 2 (eIF2), Ku proteins, and DNA-protein kinase (PK) [46,47].

In keeping with its role in viral replication and T-cell activation, NF90 was also shown to regulate the transcription of another cytokine, Interleukin 13 (IL13), by binding to a DNase I hypersensitive site (DHS) [48]. On the other hand, transcription inhibition by NF90/NF45 was observed for the major histocompatibility complex class II HLA-DR, mediated by DHS binding in B-cells [49].

More recently, NF90/NF45 was reported to be involved in the upregulation of *c-FOS* transcription upon serum induction. NF90/NF45 was shown to bind to the *c-FOS* enhancer/promoter region while cooperating with general co-activator factors [50]. Similarly, ChIP-seq data in K562 erythroleukemia cells strongly suggests that NF90/NF110, by associating with promoter regions, significantly activates the expression of transcription factors that are drivers of growth and proliferation [21]. Therefore, NF90 seems to indirectly mediate transcription regulation, with a marked implication in the immune response and cancer progression.

### 4.2. NF90 in the miRNA Biogenesis Pathway

The role of NF90 in the regulation of miRNA biogenesis was recently described. The first evidence for the involvement of NF90/NF45 in the maturation of miRNAs was shown by Sakamoto and colleagues [51]. In this study, they showed that NF90/NF45 behaves as a negative regulator of Microprocessor activity for the maturation of pri-let-7a, competing with Drosha for the binding to pri-miRNA. Therefore, the maturation of the pri-miRNA to pre-miRNA is inhibited by the binding of the NF90/NF45 complex by impairing access of Microprocessor to the pri-miRNAs.

Since this finding, different miRNAs have been shown to be modulated by NF90/NF45. The complex was found to downregulate myogenic miRNAs such as miR-133a, leading to significant loss and maturation of skeletal muscle and atrophy in NF90/NF45 double-transgenic mice [40]. More recently, NF90/NF45 was shown to inhibit the maturation of miR-7 in HCC. MiR-7 is a tumor suppressor miRNA and increased expression of NF90 leads to the inhibition of miR-7 maturation, followed by an elevated proliferation rate in HCC [52]. NF90 was therefore reported to be an oncogenic factor for HCC. Interestingly, the existence of a negative feedback loop between miR-7 and NF90 was later shown, in which mature miR-7 was able to target the 3′ UTR of NF90, leading to its translational repression [53].

It was recently demonstrated that NF90/NF45 is able to inhibit the maturation of miR-3173, a miRNA embedded in the first intron of Dicer pre-mRNA, by preventing binding of Microprocessor to pri-miR-3173. Furthermore, in the absence of NF90, the level of pre-miR-3173 increases while Dicer pre-mRNA exhibits splicing defects that lead to its downregulation. Increased progression and metastasis were observed in ovarian cancer cells. Therefore, it was established that NF90 can act as a tumor suppressor in ovarian cancer models. Interestingly, the mature form of miR-3173 is able to target NF90 mRNA by binding to its 3′ UTR, leading to translational repression, mediating a feedback amplification loop that controls Dicer expression and ovarian carcinoma progression [22].

More recently, the extent of the effect of NF90 on miRNA maturation was uncovered using genome-wide approaches [27]. Data indicate that NF90 is able to directly bind and modulate the processing of a specific subset of human miRNA precursors that are weakly bound by Microprocessor, suggesting that NF90 and Microprocessor might be in competition for the binding of pri-miRNAs. Moreover, in agreement with other studies [20], it was found that NF90-bound and modulated pri-miRNAs are highly stable RNA structures, displaying significantly longer duplexes with fewer and smaller bulges compared to all human pri-miRNAs [27].

Interestingly, transcriptomic analysis showed that NF90 association with pri-miRNAs may modulate the expression of their host genes, as described previously [22]. For instance, the expression of T-Lymphoma Invasion and Metastasis-Inducing Protein 2 (TIAM2), hosting pri-miR-1273c, is downregulated upon the loss of NF90 [27]. TIAM2 is a well-known oncogene and metastasis factor in HCC and its overexpression has been shown to promote proliferation and invasion in liver cancer [54]. Along the same lines, several pri-miRNAs that are bound and modulated by NF90 such as miR-34a are tumor-suppressor miRNAs that are downregulated during cancer development and progression [55]. NF90 binding to these pri-miRNAs would prevent their processing by Microprocessor, thereby promoting cancer progression. In addition to mir-34a, NF90 inhibits the biogenesis of several other miRNAs with known tumor suppressor activities such as miR-16, miR-128, and miR-145 [27,56]. Interestingly, in bladder cancer, miR-145 was shown to be part of a dysregulated signaling axis involving lncRNA-LET and NF90, which is responsible for drug resistance. In particular, gemcitabine treatment was shown to directly repress lncRNA-LET, leading to higher NF90 stability, which in turn inhibited the maturation of miR-145 and enhanced the accumulation of chemotherapy-induced cancer stem-like cells [44].

On the other hand, NF90 can also bind and stabilize oncogenic pri-miRNAs such as pri-miR-548k, which is implicated in esophageal squamous cell carcinoma (ESCC), promoting cancer progression, metastasis, and poor overall survival [42,57]. Interestingly, miR-548k was shown to upregulate NF90 and downregulate lncRNA-LET, which acts as a tumor suppressor, establishing a feedback amplification loop that controls ESCC progression and metastasis [42,57].

It is not known whether NF45 is also implicated in the regulation of pri-miRNA processing, together with NF90. The stable interaction between these two factors might suggest that NF45 is likely to be involved, although it has not yet been formally demonstrated. In addition, most studies suggest that NF90 regulates pri-miRNA processing via competition with Microprocessor for binding to pri-miRNA structures. However, a more direct functional role of NF90 in inhibiting pri-miRNA processing has not been formally excluded.

To conclude, the effect of NF90 on cancer development and progression is likely multifactorial, being dictated not only by the effect of the miRNAs it modulates, but also by their mRNA targets, and the gene that hosts the pri-miRNA. Therefore, the outcome of differential NF90 expression in cancer depends on complex mechanisms that warrant further investigation.

### 4.3. NF90 in mRNA Translation, Stability, and Degradation

In addition to controlling mRNA fate by binding and modulating the processing of miRNAs, NF90 can directly regulate mRNAs’ translation, stability, and decay [19,58]. AU-rich regions are frequently found in 3′ UTRs of mRNAs and their recognition by RBPs often determines their fate [3,59]. Ribonucleoprotein immunoprecipitation (RIP) analysis showed that NF90 is able to bind an AU-rich signature motif, known as NF90m, found in a large subset of mRNAs [39]. However, the consequence of NF90 binding can vary depending on the target mRNA or on the condition studied. For example, NF90 binding to 3′ UTRs is able to regulate the stability of mRNAs and their translation, either positively or negatively [16,23,39,58,60].

While NF90 has been shown to modulate the translation of bound mRNAs, its effect can be either positive or negative, depending on the RNA target. NF90 can inhibit translation by affecting the initiation step, or by retaining target mRNAs in the nucleus [8]. For instance, insertion of the AU-rich NF90m into a reporter gene did not affect mRNA stability but rather inhibited its translation by preventing its association with actively translating ribosomes [39].

NF90 has also been shown to activate the translation of a limited subset of mRNAs, such as Vascular Endothelial Growth Factor (VEGF) and cyclin T1 mRNA [17,61]. Under hypoxic conditions, NF90 is able to interact with the 3′ UTR stem-loop hypoxia stability region in *VEGF* mRNA, promoting its loading onto polysomes and increasing its stability [19]. After Human Immunodeficiency Virus 1 (HIV1) infection, NF90 promotes viral replication and latency by binding to cyclin T1 mRNA 3′ UTR and facilitating the recruitment of translation initiation factors [62].

NF90 is also able to increase or decrease the stability of its target mRNAs. For instance, upon co-stimulation of CD28 following T-cell activation, NF90 is phosphorylated by AKT and translocates to the cytoplasm where it can bind the 3′ UTR of *IL2* mRNA, which prevents its degradation [32,63]. Similarly, it was shown that NF90, after its translocation to the cytoplasm mediated by CDK2 phosphorylation, promotes HCC proliferation by stabilizing cyclin E1 mRNA [23,61]. In HCC, NF90 was also shown to regulate the stability of Poly(ADP-Ribose) Polymerase 1 (*PARP1*) mRNA, and therefore promote tumor development [60]. On the other hand, NF90 was shown to destabilize viral mRNAs, such as vesicular stomatitis virus (VSV) mRNAs in the cytoplasm upon VSV infection [16].

More recently, several studies have suggested the existence of signaling pathways or regulatory axes involving NF90, and often, miRNAs that impact the development of pathologies, especially cancer progression and metastasis [18,37,42,44]. For instance, NF90 was shown to suppress the translation of cytokine B-cell activating factor (*BAFF*) mRNA by recruiting miR-15a to *BAFF* 3′ UTR. Moreover, a variant of *BAFF* mRNA (*BAFF*-*var*), which lacks the NF90 binding site, was found to have enhanced translation and is associated with an elevated risk of the development of autoimmune diseases. Interestingly, it was shown that the binding of NF90 and miR-15 to the mRNA correlates with higher Ago2 binding, which suggests that translational inhibition of *BAFF* mRNA might be mediated by RISC [58].

Interestingly, NF90 has been identified as a subunit of P-bodies, and was recently shown to interact with RBPs involved in RISC-mediated silencing, such as Moloney leukemia virus 10 (MOV10) and Argonaute 2 (AGO2), in an RNA-dependent manner [26,64]. NF90 and MOV10 can bind a common set of target mRNAs. However, RIP analyses suggest that the binding of MOV10 to target mRNAs might prevent or reduce the binding of NF90 to the same targets, and vice versa [26]. Interestingly, the loss of NF90 increases the association of AGO2 to the target mRNAs while reducing their abundance. These findings suggest that NF90 might have a role in RISC-mediated silencing by modulating AGO2 association with target mRNAs, and thereby enhancing their stability [26]. Similarly, the heterodimer NF90/NF45 was recently shown to compete with factors involved in Staufen-mediated mRNA decay (SMD) [65]. In particular, NF90/NF45 competes with the SDM machinery for the binding of 3′ UTRs of a subset of mitotic mRNAs, leading to their stabilization and increased expression [65]. Despite the numerous examples of mRNAs regulated by NF90 at the post-transcriptional level, the exact mechanism of inhibition or the activation of translation or mRNA stabilization is still largely unknown. However, recent findings suggest that the function of NF90 in mRNA translational regulation and stability might be mediated by RISC and SMD activity [26,65].

### 4.4. NF90 in Viral Replication

Besides acting as a cellular mRNA binding factor that controls RNA metabolism and translation, NF90 also binds viral RNA or DNA [66]. The consequence of its binding can vary, supporting or inhibiting viral replication and viral genome expression, depending on the type of virus [29]. Numerous viruses have been shown to exploit NF90 to support their replication such as hepatitis C virus (HCV), HIV, human papilloma virus (HPV), and Dengue virus (DV) [43,67,68,69]. For instance, NF90 was shown to bind the 5′-terminal sequence of the HCV RNA genome upon infection and promote HCV replication by possibly associating with the replication complex [69]. Upon HIV infection, NF90 shows a pleiotropic effect by stimulating the viral gene expression as well as stabilizing HIV RNA [68]. On the other hand, NF90 is able to act as a host antiviral factor for other types of viruses such as the influenza A virus (IAV) and Ebola virus (EBOV) [70]. For example, NF90 was found to suppress EBOV replication by associating with Viral Protein 35 (VP35) and impairing the function of EBOV replication complex [16,70]. Moreover, it was recently shown that viral infection promotes the translocation of NF90 from the nucleus to the cytoplasm. NF90 in the cytoplasm is able to bind viral mRNAs, leading to the inhibition of viral infection [16]. Similarly, following viral infection, the interferon-inducible kinase, PKR, phosphorylates NF90, leading to its dissociation from NF45 and export from the nucleus [33,71]. Phosphorylated NF90 accumulates on ribosomes where it associates with viral RNA, inhibiting their translation [33].

Despite the numerous and diverse examples of NF90 activity during the response to viral infection, the exact mechanism underlying its role is yet to be fully elucidated. However, it is possible that the complicated contribution of NF90 to antiviral immunity might occur through different mechanisms depending on the type of virus.

### 4.5. Interplay between NF90 and Circular RNAs

The most recent described role of NF90 involves circular RNA (circRNA) biology. In fact, it was found that NF90 is a key factor for the biogenesis of circRNAs in the nucleus while their mature form might act as a molecular reservoir of NF90 in the cytoplasm for prompt immune response following viral infection [16]. In particular, in the nucleus, NF90 promotes back-splicing of circRNAs by binding to flanking introns of circularized exons and stabilizing the transient RNA duplexes that juxtapose the splice sites [72]. Interestingly, upon viral infection, the nuclear pool of NF90 translocates into the cytoplasm, reducing the formation of circRNAs in the nucleus. On the other hand, the re-distribution of NF90 in the cytoplasm increases its binding to circular ribonucleoprotein (circRNP) complexes that compete with viral RNAs for the binding of NF90 [16].

It was recently shown that circular Actin Alpha 2 (circACTA2) RNA can compete with CDK4 mRNA for the binding of NF90 in vascular smooth muscle cells (VSMC). In particular, increased circACTA2 expression, induced by angiotensin II (Ang II) stimulation in VSMC, reduced the association of NF90 with CDK4 mRNA, leading to Ang II mediated senescence in VSMC [73]. Although this newly discovered function of NF90 is not yet supported by extensive literature, the regulatory potential of circRNAs and NF90 highlights the significance of this mechanism and encourages further investigation.

## 5. Conclusions

Since its identification, NF90 has been studied with increasing interest, especially in the last 15 years. Recently, a growing number of functions of NF90 have been described that allow us to define it as a polyvalent and versatile factor in RNA biology and homeostasis. In fact, NF90 has been shown to be implicated in RNA metabolism at all stages, from transcription to degradation. Moreover, its role in miRNA biogenesis, RNA stability, and translation during cancer progression and viral infection highlights its ability to orchestrate cellular responses in different cellular compartments and upon different stimuli. Given the widespread impact of the binding of NF90 to RNAs during homeostasis, viral infection, and cancer progression, it would be tempting to envision the application of this knowledge for the development of therapeutics designed in order to inhibit or promote NF90 binding to selected RNAs. Many NF90-modulated pri-miRNAs are associated with cancer development and progression. Therefore, the modulation of NF90 binding to these pri-miRNAs would affect their abundance, having an effect not only on their mature form but also on the expression of their tumor-related mRNA targets. Moreover, the effect of the competition between Microprocessor and NF90 would in turn also affect the expression of the mRNAs hosting the pri-miRNA, which can be tumor suppressors such as Dicer or oncogenes such as TIAM2. NF90 was recently shown to compete with SMD and RISC activities, leading to increased stability of NF90-bound mRNAs that can be involved in mitosis, viral infection, and hypoxia. A better understanding of how NF90 is implicated in RISC and SMD-mediated silencing might clarify its role in mRNA translation and degradation during viral infection or cancer-induced hypoxia, and potentially open possibilities for targeted therapies. Interfering with NF90 binding to pri-miRNAs or mRNAs using nanotherapeutics could be a novel approach to control the abundance of specific targets in cancer.

## Figures and Tables

**Figure 1 ijms-23-13584-f001:**
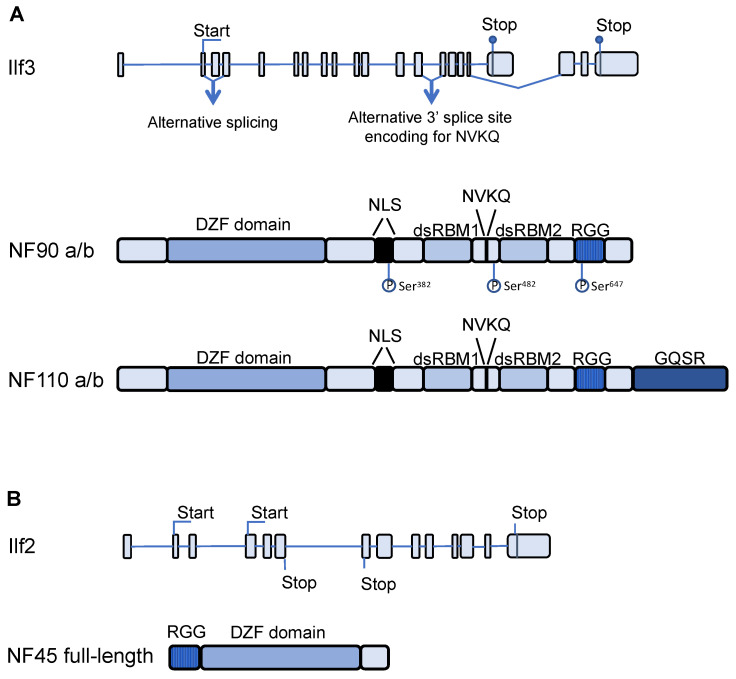
Schematic representation of *ILF3* and *ILF2* transcripts and domain structure of NF90, NF110, and NF45. (**A**). Isoforms of transcripts derived from the *ILF3* gene showing alternative splicing sites and start and stop codons. Protein products of the *ILF3* transcripts, NF90a/b and NF110a/b. Phosphorylation sites on NF90 are indicated. (**B**). Transcript derived from *ILF2* gene showing start and stop codons. Protein product of *ILF2* transcript, NF45. DZF indicates a zinc finger associated domain, NLS indicates nuclear localization signal, dsRBM1 indicates double stranded RNA binding motif 1, NVKQ indicates Asparagine-Valine-Lysine-Glutamine, dsRBM2 indicates double stranded RNA binding motif 2, RGG indicates Arginine-Glycine-Glycine repeats, and GQSR indicates the Glycine-Glutamine-Serine-Arginine motif.

**Figure 2 ijms-23-13584-f002:**
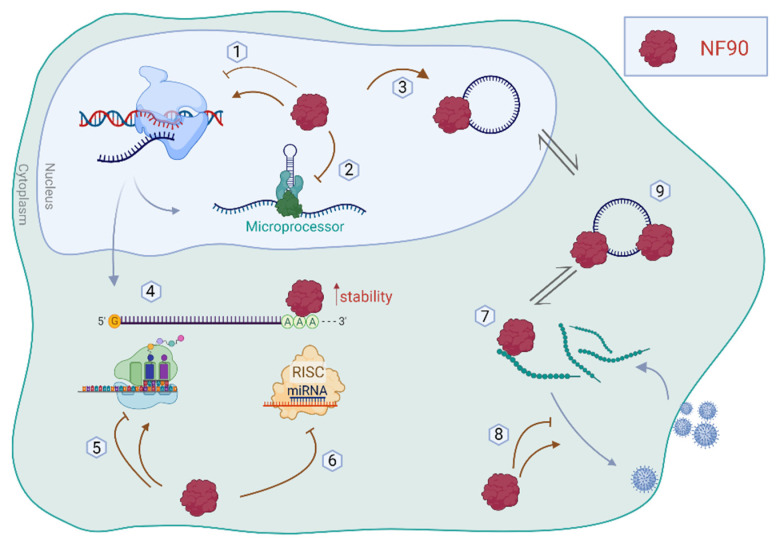
Schematic representation of the functions of NF90 in the cytoplasm and nucleus. NF90 is involved in (1) transcription regulation, (2) miRNA biogenesis, and (3) circRNA biogenesis, in the nucleus. In the cytoplasm, NF90 can regulate (4) mRNA stability, (5) translation of mRNA, and (6) RISC-mediated silencing. In addition, NF90 is implicated in viral infection by (7,8) binding viral genome/proteins. (9) NF90 can be sponged by circRNAs, modulating its availability in the cytoplasm.

## Data Availability

Not applicable.

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
