# Peer review of "The Polyvalent Role of NF90 in RNA Biology"

_ijms, 2022, doi:10.3390/ijms232113584_

Round 1

Reviewer 1 Report

Nuclear Factor 90 (NF90) is a multifunctional DNA- and RNA-binding protein found mainly in the nucleus and encoded by the interleukin enhancer-binding factor 3 (ILF3) gene. NP90 is involved in transcription, translation, mRNA export, RNA splicing, mRNA turnover and microRNA biogenesis, and can act as cellular partners affecting viral replication and host defense. This review summarizes various functions of NP90.

The manuscript summarizes a lot of information and satisfactorily reviews the recent literature on NP90. Nevertheless, the manuscript is not organized at its best, and some aspects may be further developed to gain in clarity and completeness. Therefore, I suggest the following changes:

(1) It would be helpful to show a diagram of ILF3 gene in Figure 1, as well as NF90 and NF110 mRNA slicing isoforms. Since the authors mention NF90a/b and NF110a/b in the main text (lines 64–65), I also suggest labelling an additional alternate splicing event at the competing 3′ splice site of the intron 13/exon 14 boundary, which inserts four amino acids (NVKQ) into the region between the two dsRBDs.

(2) The reader does not get a clear idea about binding preferences of NF90.

First, lines 77–79 read, “This analogy suggests that, like for ADAR2, a sequence motif in the target dsRNA might influence NF90 binding. In contrast with this observation, it was shown that NF90 is able to recognize dsRNAs exclusively based on their structure.” I don’t think that these observations are contradictory. In the crystal structure with a synthetic dsRNA, tandem dsRBDs of NF90 recognize a double-stranded helical structure. Base-specific interactions with a G-X10-A motif in the minor groove stabilize binding further. It would be worth mentioning this preferred motif in the main text. Does NF90 bind its mRNA target via this motif, or it does not exist in transcriptome?

Second, lines 245–247 read, “Ribonucleoprotein immunoprecipitation (RIP) analysis showed that NF90 is able to bind an AU-rich, 25-to-30 nucleotide signature motif, known as NF90m, found in a large subset of mRNAs.” This does not agree with the crystal structure and needs to be discussed.

(3) Section “2. NF90 structure and RNA binding mode” is a perfect example of a suboptimal organization of the manuscript. Information relevant to NF90 binding preferences is in the section 2 but can be also found in other sections of the manuscript, making it more challenging for the reader to get a complete picture. This also applies to other sections.

(4) I would first describe what NF90 does, and then how NF90 itself is regulated. In other words, I would swap the order of sections 3 and 4.

Reviewer 2 Report

This manuscript focuses on the expression and function of RNA binding protein, NF90. Overall, the manuscript is well written and organized. It provides a good foundation of knowledge with respect to NF90, specifically focusing on it's role in RNA biology. A few specific comments:  

It would be helpful to include a bit more genomic information to section 2, perhaps the genomic location of Ilf3 gene, info on the promoter, and the number of isoforms.  

It would be helpful to include NF45 in figure 1 along with NF90 and NF110. Perhaps even phosphorylation sites?

With respect to miRNA biogenesis, does the NF90/45 complex simply compete with Dicer for pri-miRNA processing or does it have a functional effect?

Additional discussion of the function of NF90 when complexed with NF45 vs. the function of NF90 independently would be helpful. Particularly with respect to miRNA biogenesis. What what determines whether NF90 acts independently or in a complex? Is NF90 of NF45 expression rate limiting?

Figure 2 is a bit hard to read. 

Round 2

Reviewer 1 Report

The Authors have addressed all of my concerns with the original manuscript. The revised manuscript is ready for publication.